# Magnetic and Magnetostrictive Behaviors of Laves-Phase Rare-Earth—Transition-Metal Compounds Tb_1−x_Dy_x_Co_1.95_

**DOI:** 10.3390/ma15113884

**Published:** 2022-05-29

**Authors:** Chao Zhou, Kaili Li, Yuanliang Chen, Zhiyong Dai, Yu Wang, Liqun Wang, Yoshitaka Matsushita, Yin Zhang, Wenliang Zuo, Fanghua Tian, Adil Murtaza, Sen Yang

**Affiliations:** 1School of Physics, MOE Key Laboratory for Nonequilibrium Synthesis and Modulation of Condensed Matter, Xi’an Jiaotong University, Xi’an 710049, China; likaaili2013@stu.xjtu.edu.cn (K.L.); xajtcyl@stu.xjtu.edu.cn (Y.C.); daizhiyong@stu.xjtu.edu.cn (Z.D.); yuwang@xjtu.edu.cn (Y.W.); wanglq@xjtu.edu.cn (L.W.); yzhang18@xjtu.edu.cn (Y.Z.); zuowenliang@xjtu.edu.cn (W.Z.); tfh2017@xjtu.edu.cn (F.T.); adilmurtaza91@xjtu.edu.cn (A.M.); 2National Institute for Materials Science, Tsukuba 305-0047, Ibaraki, Japan; matsushita.yoshitaka@nims.go.jp

**Keywords:** morphotropic phase boundary, magnetostriction, Laves-phase, temperature dependence, composition dependence

## Abstract

The magnetic morphotropic phase boundary (MPB) was first discovered in Laves-phase magnetoelastic system Tb–Dy–Co alloys (PRL 104, 197201 (2010)). However, the composition-dependent and temperature-dependent magnetostrictive behavior for this system, which is crucial to both practical application and the understanding of transitions across the MPB, is still lacking. In this work, the composition-dependence and temperature-dependence of magnetostriction for Tb_1−x_Dy_x_Co_1.95_ (x = 0.3~0.8) are presented. In a ferrimagnetic state (as selected 100 K in the present work), the near-MPB compositions x = 0.6 and 0.7, exhibit the largest saturation magnetization M_S_ and the lowest coercive field H_C_; by contrast, the off-MPB composition x = 0.5, exhibits the largest magnetostriction, the lowest M_S,_ and the largest H_C_. Besides, a sign change of magnetostriction is observed, which occurs with the magnetic transition across the MPB. Our results suggest the combining effect from the lattice strain induced from structure phase transition, and the influence of the MPB on magnetocrystalline anisotropy. This work may stimulate the research interests on the transition behavior around the MPB and its relationship with physical properties, and also provide guidance in designing high-performance magnetostrictive materials for practical applications.

## 1. Introduction

Magnetostrictive materials can realize the conversion between magnetic energy and mechanical energy, thus they are widely used in the key components of sensors, actuators, and transducers [1]. Due to the 3d–4f exchange interactions [2], many Laves-phase rare-earth-transition-metal compounds (RTx, R denotes the rare-earth elements and T denotes the transition-metal elements, x = 1.8~2) exhibit giant magnetostriction, e.g., TbFe_2_, DyFe_2_, TbCo_2_, and DyCo_2_ [3,4].

In 2010, Yang et al. reported the magnetic morphotropic phase boundary (MPB) in a Laves-phase pseudo-binary Tb_1−x_Dy_x_Co_2_ system [5]. Later on, Bergstrom et al. reported the MPB in the classic system-Terfenol-D [6]. Afterwards, MPBs were discovered in more and more Laves-phase systems such as Tb–Gd–Co, Tb–Gd–Fe, Tb–Nd–Co, etc. [7,8,9]. The transition at the MPB involves not only the change of magnetic ordering, but also the change of structural ordering, thus yielding exotic magnetoelastic and magnetocaloric properties [10,11,12,13].

In practical applications, temperature variation is inevitable and impact on the properties of the magnetostrictive materials, so will influence the performance of the devices (sensors, actuators, and transducers). Although MPB has been utilized to realize large magnetostriction, it is not only composition-dependent but also temperature-dependent. Whether one magnetostrictive material fits for an application, is determined not only by its maximum magnetostriction, but also by the temperature-dependent performance [14]. In view of this point, the MPB composition might not be the best candidate for application.

For a certain magnetostrictive material system, to optimize the composition to meet the application requirements, it is necessary to acquire both the temperature-dependence and composition-dependence of magnetostriction. For the proto magnetic MPB system-Tb–Dy–Co alloys, despite much intense research, e.g., spin reorientation behavior [15], magnetocaloric effect [16], the nature of transition [17], and the role of the electronic structure based on first principle calculation [18], investigation on the temperature-dependence and composition-dependence of magnetostriction, as well as the relation between the changing trend with MPB, is still lacking. Thus, we proposed to clarify the composition and temperature-dependent magnetostrictive behaviors in the MPB-involved systems.

In this work, for the Tb–Dy–Co system, the magnetostrictive behavior as a function of temperature and composition were investigated and discussed regarding the composition-dependence of magnetic properties, e.g., magnetization, coercive field, and magnetic susceptibility.

## 2. Materials and Methods

It is noticed that reducing the content of transition metal favors the formation of the Laves-phase compound [19,20,21], so the formula of stoichiometry is fixed to be Tb_1−x_Dy_x_Co_1.95_ (x = 0.3~0.8). The Tb_1−x_Dy_x_Co_1.95_ alloy samples were prepared by arc melting method with the raw materials of Tb (99.9%), Dy (99.9%), and Co (99.9%) in argon atmosphere. In order to guarantee the composition homogeneity, the magnetic stirring was employed during the arc melting process and all ingots were melted six times. Samples used for physical property measurements are polycrystalline. The crystal structure was examined by X-ray diffraction (XRD, Bruker D8 ADVANCE, Hamburg, Germany) using Cu Kα radiation (λ = 0.154056 nm) with an angle (2θ) step of 0.02° between 20° and 80°. The line scans of chemical elements were performed by scanning electron microscopy (SEM, JSM-7000F, JEOL, Tokyo, Japan). The chemical compositions were analyzed using X-ray fluorescence (XRF, Bruker S8 Tiger, Hamburg, Germany). The magnetization (*M*) versus magnetic field (*H*) hysteresis loops, and the magnetic susceptibility (*χ*) versus temperature (*T*) curves were measured using superconducting quantum interference device (SQUID, Quantum Design, Santa Barbara, CA, USA). The magnetostriction at the field of 20 kOe from 10 K to 130 K, was measured by the standard strain gauge technique with a gauge factor of 2.11 ± 1%, combined with the temperature controlling system (Cryostat Device, Cambridge, UK).

## 3. Results and Discussion

### 3.1. Crystal Structure Characterization

The XRD profiles for the selected compositions (x = 0.3~0.8) at room temperature (~298 K) are shown in Figure 1a. All of the samples possess a pure C15 cubic Laves-phase structure (space group Fd$\overset{—}{3}$m) [22], without any second phase (RT_3_) that usually appears in Laves-phase rare-earth-transition metal alloys [23,24,25]. Figure 1b plots the corresponding crystal structure. It should be noted that when the temperature is below the Curie temperature T_C_, the non-cubic structure symmetry can be detected using neutron diffraction or synchrotron XRD [5,26,27,28].

The line scans of elements using SEM (Appendix A), suggest the compositional homogeneity for all the available samples. It is also necessary to point out that the microstructure, i.e., the grain size, of Laves-phase intermetallic compounds usually does not play the key role for magnetic properties [29]. The comparison between the chemical relative mass percentages from the experiment and calculation (XRF results; Appendix A) reveals the consistence as the content of Dy increases. The deviation might stem from the loss in the arc-melting procedure.

### 3.2. Temperature Spectrum of Magnetic Susceptibility and Magnetic Phase Diagram

The susceptibility versus temperature curves χ-T are shown in Figure 2(a1–a6) (the inverse susceptibility versus temperature curves 1/χ-T can be referred to in Appendix A). Tb-rich composition (i.e., x = 0.3) and Dy-rich composition (i.e., x = 0.8) exhibit only one peak that denotes the paramagnetic–ferrimagnetic phase transition. The samples of the intermediate composition range (i.e., x = 0.4~0.7) show two peaks, of which the one appearing at the higher temperature indicates the paramagnetic–ferrimagnetic transition and the one appearing at the lower temperature indicates the ferrimagnetic–ferrimagnetic transition. As proposed from the previous research work [5,27,30], the easy magnetization axis (EMA) of Tb-rich compositions aligns along [111], while that of the Dy-rich compositions aligns along [001]. Such a ferrimagnetic–ferrimagnetic transition is coined as the spin reorientation transition (SRT) [31].

Based on the EMA for two end members and the phase transition temperatures determined from the χ-T curves, the magnetic phase diagram is illustrated in Figure 2b. It can be seen that the MPB derives from the triple-point (the intersection point of the T_C_ line and the ferrimagnetic (EMA//[111])-ferrimagnetic (EMA//[001]) phase boundary). Since the magnetocrystalline anisotropic coefficient K1 values of Tb^3+^ ions and Dy^3+^ ions are negative and positive [32], respectively, the spin reorientation transition temperature T_SRT_ depends on the Tb/Dy ratio. With the increasing content of Dy, the T_C_ decreases while the T_SRT_ increases; the former is attributed to the less strength of the 3d–4f coupling between Dy and Co [33], and the latter is attributed to the enhanced 3d–4f–5d hybridization [34].

### 3.3. M-H Hysteresis Loops

To study the composition dependence of magnetic properties across the MPB, the measurement temperature is usually fixed at below the T_C_ [5], i.e., 100 K at the present work. The *M*–*H* hysteresis loops at 100 K are shown in Figure 3(a1–a6). The composition dependence of coercive field H_C_ and the saturation magnetization M_S_ (calculated using the law of approach to saturation [35]) are shown in Figure 3(b1,b2).

The magnetic properties of Laves-phase rare earth—transition metal compounds, are dominated by the highly anisotropic rare earth sublattice, especially the distortion of the spherical 4f charge density of the rare earth sublattice [36]. As for the Tb–Dy–Co system, at a certain temperature, the anisotropic coefficient is definitely composition-dependent. Because of the differences of the 4f electron configurations in Tb^3+^ (4f^8^) and Dy^3+^ (4f^9^), the magnetic properties (M_S_, H_C_, etc.) exhibit a competition effect from both the Tb-sublattice and the Dy-sublattice. The compositions x = 0.6 and 0.7, both locating close to the MPB (where the compensation of anisotropy occurs at T_SRT_), exhibit a large M_S_ and a low H_C_, which is attributed to the facility of magnetic domain switching resulted from the low magneto-crystalline anisotropy and low energy barrier at the MPB [37,38]. By contrast, the composition x = 0.5, which locates further to the MPB than x = 0.6 and 0.7, shows the lowest M_S_ and the largest H_C_, reflecting the competition effect from both the Tb-sublattice and the Dy-sublattice that reaches the extreme at x = 0.5. For 0.3 and 0.4, the Tb-sublattice is believed to play the dominant role.

### 3.4. Composition- and Temperature-Dependent Magnetostriction

Figure 4a shows the magnetostriction (ε) curves of x = 0.3~0.8 in the temperature range from 10 K to 130 K. For x = 0.3, 0.4, and 0.5, ε remains positive and increases monotonously with the decrease in temperature. For x = 0.6, 0.7, and 0.8, ε exhibits positive values at higher temperatures and negative values at lower temperatures. The T_SRT_ of x = 0.3 is ~6 K (the signal is too weak so that it cannot be clearly seen on Figure 2(a1)), out of the temperature region of 10 K~130 K. Within this temperature region, x = 0.3 possesses a rhombohedral phase (EMA//[111]), so exhibits positive magnetostriction.

Given that the transitions of Laves-phase intermetallics involve not only magnetic ordering but structural change, the magnetostriction, which originates from magnetoelastic coupling, can be well interpreted using a domain switching mechanism [5,27]. Based on the model proposed by Yang et al. [27], the magnetostriction is proportional to the lattice strain, which arises from the structural transition of magnetic materials. And conversely, the sign change of the measured magnetostriction indicates the change of crystal structure symmetry [14]. Moreover, the crystal structure symmetry conforms to the spontaneous magnetization M_S_ direction (consistent with the EMA) [27]. Therefore, the sign change of ε for x = 0.6, 0.7 and 0.8 demonstrates both magnetic transition between two different ferrimagnetic phases (EMA//[111] and EMA//[001]), and concurring structure transition between rhombohedral and tetragonal phases [14,27]. It should be paid attention to that, the T_SRT_ of x = 0.4 and 0.5 lie within the temperature range of 10 K~130 K (Figure 2b), but their magnetostrictions remain positive. This may be ascribed to the transition route at the MPB under the large external magnetic field, which is determined by the degree of magnetic ordering of two end members that form magnetic MPB [9].

Figure 4b shows the contour diagram of ε as a function of composition and temperature. For comparison, the magnetic properties of the current system are compared with those of the previously reported Tb_1−x_Dy_x_Co_2_ system, as shown in Table 1. The composition-dependent sign change of the magnetostriction, together with the phase diagram (Figure 2), suggest that ε is influenced mainly by two factors: (1) the facility of magnetic domain switching as discussed above and (2) the theoretical saturated strain that is determined by the lattice strain [27]. The composition x = 0.5, located further away from the MPB (where the magneto-crystalline anisotropy of two end members compensate) [14], possesses a larger magneto-crystalline anisotropy than x = 0.6 and 0.7. Meanwhile, x = 0.5 locates closer to the end member of TbCo_2_, so it possesses a larger lattice strain upon domain switching, which results from the switching of the distorted rhombohedral lattice [27]. This observation is consistent with that reported in another MPB-involved system Tb_1−x_Dy_x_Fe_2_ [39]. Interestingly, such a phenomenon was also observed recently in a ferroelectric MPB-involved system [40].

## 4. Conclusions

In conclusion, the magnetic and magnetostrictive properties of Tb_1−x_Dy_x_Co_1.95_ (x = 0.3~0.8) alloys were systematically studied. The results reveal that the magnetic properties (M_S_ and H_C_) are strongly influenced by the MPB, while the magneto-elastic property (ε) relies mainly on the composition-dependent crystal lattice distortion. Therefore, the off-MPB composition with EMA//[111], i.e., x = 0.5, exhibits the largest ε, the largest H_C,_ and the lowest M_S_; by contrast, the near-MPB compositions x = 0.6 and 0.7 exhibit the largest M_S_ and the lowest H_C_, as well as a lower ε, compared with x = 0.5 and other Tb-rich compositions. Our work demonstrates the temperature-dependence and composition-dependence of magnetostriction for the proto magnetic MPB system Tb–Dy–Co and may accelerate the design of optimum magnetostrictive materials for energy conversion devices.

## Figures and Tables

**Figure 1 materials-15-03884-f001:**
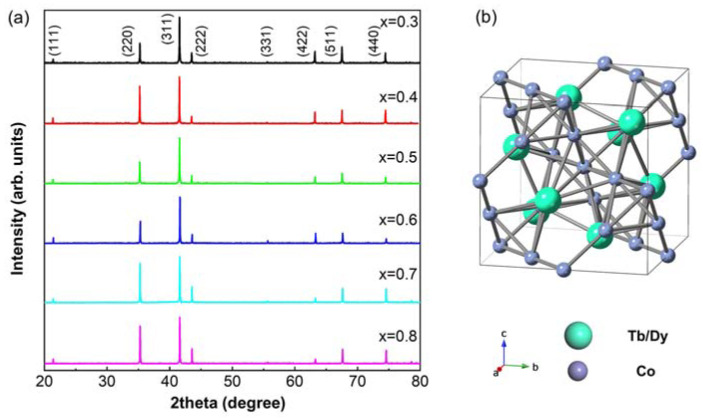
(**a**) X-ray diffraction profiles of Tb_1−x_Dy_x_Co_1.95_ alloys (x = 0.3, 0.4, 0.5, 0.6, 0.7, 0.8), (**b**) the crystal structure of Laves-phase Tb_1−x_Dy_x_Co_1.95_ alloys.

**Figure 2 materials-15-03884-f002:**
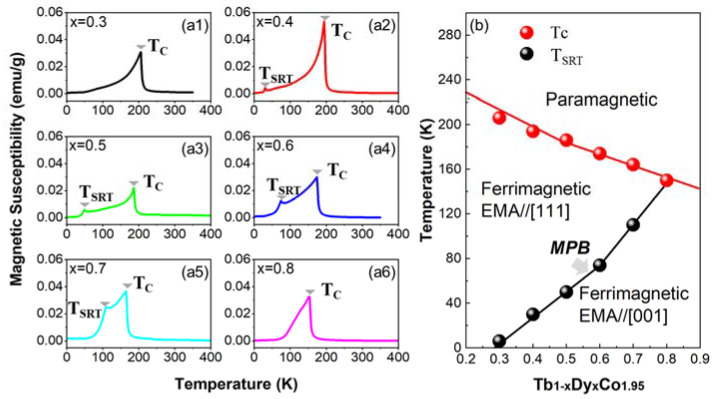
(**a1**–**a6**) Magnetic susceptibility versus temperature curves and (**b**) the phase diagram of Tb_1−x_Dy_x_Co_1.95_.

**Figure 3 materials-15-03884-f003:**
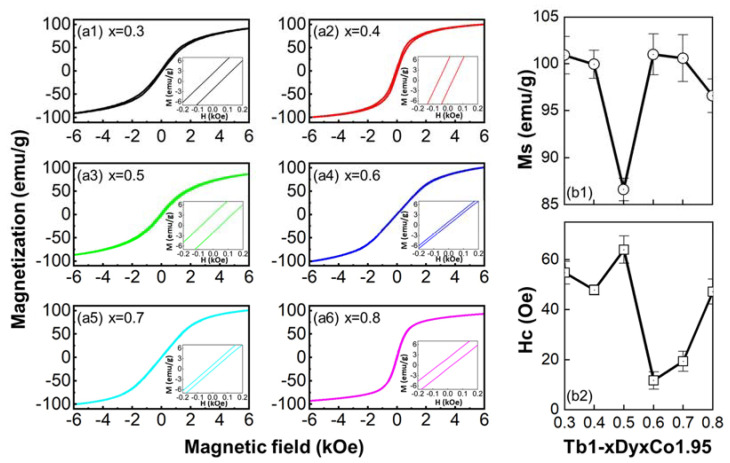
(**a1**–**a6**) Magnetization (*M*) versus Magnetic field (*H*) hysteresis loops of Tb_1−x_Dy_x_Co_1.95_ alloys (x = 0.3, 0.4, 0.5, 0.6, 0.7, 0.8); Composition dependence of saturation magnetization M_S_ (**b1**) and coercive field H_C_ (**b2**) at 100 K.

**Figure 4 materials-15-03884-f004:**
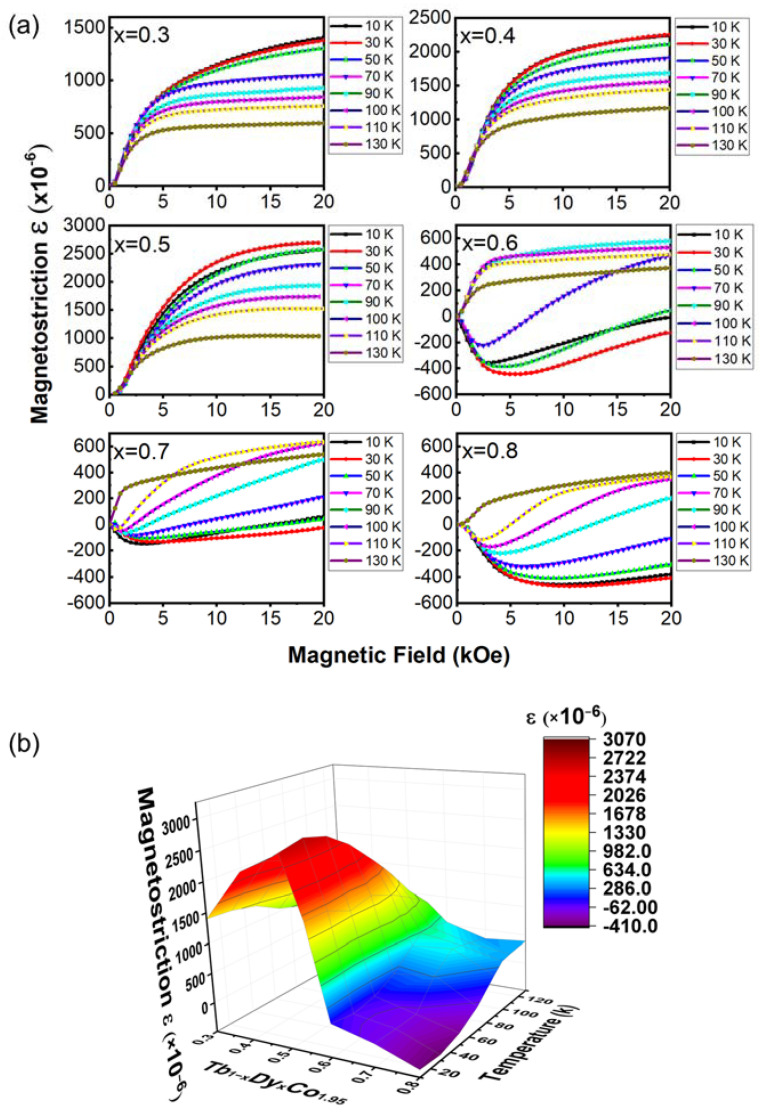
(**a**) Temperature-dependent magnetostriction curves of Tb_1−x_Dy_x_Co_1.95_ alloys (x = 0.3, 0.4, 0.5, 0.6, 0.7, 0.8), (**b**) 3-dimensional diagrams of the magnetostriction as a function of composition and temperature.

**Table 1 materials-15-03884-t001:** Comparison of magnetic properties between selected compositions of Tb_1−x_Dy_x_Co_1.95_ and Tb_1−x_Dy_x_Co_2_ systems.

Composition	M_S_ (emu/g)	H_C_ (Oe)	ε (ppm) at 110 K	Figure of Merit ׀ε׀/H_C_ (Oe^−1^·10^6^)	Reference
Tb_0.6_Dy_0.4_Co_1.95_	99.9	47.9	1441	30.1	The present study
Tb_0.5_Dy_0.5_Co_1.95_	86.6	64	1523	23.8	The present study
Tb_0.4_Dy_0.6_Co_1.95_	101.1	11.7	475	40.6	The present study
Tb_0.3_Dy_0.7_Co_1.95_	100.6	19.4	635	32.7	The present study
Tb_0.6_Dy_0.4_Co_2_	115	102	1410	13.8	Ref. [5]
Tb_0.3_Dy_0.7_Co_2_	105.5	15.6	828	53.1	Ref. [5]

## Data Availability

Not applicable.

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
