# Peer review of "Magnetic and Magnetostrictive Behaviors of Laves-Phase Rare-Earth—Transition-Metal Compounds Tb1−xDyxCo1.95"

_materials, 2022, doi:10.3390/ma15113884_

Round 1

Reviewer 1 Report

The authors present the results in a very concise way, there is no deeper interpretation of them. Despite this, the work is suitable for publication as a report of original results obtained from magnetic and magnetostriction investigations. It is a pity that no structural tests have been carried out in the temperature range where phase changes, both magnetic and structural, occur.
Some minor criticism: In Figure 3a(1-6) one can not see the hysteresis loops mentioned in the text, which probably are observed in the extended  M-scale.

Reviewer 2 Report

The authors present structural and magnetic measurements on the Laves phase series Tb1-xDyxCo1.95 (x = 0.3 to 0.8).

Especially, they are interested on the magnetostriction. From magnetic susceptibility measurements they find two transition temperatures: TC between para- and ferrimagnet (with 111 easy axis direction and TSRT between this ferrimagnetic phase and a second ferrimagnet with 001 easy axis. XRD- results are only used to proof phase purity of the samples. The interpretation of the three magnetic phases is not proofed by analyses of the magnetic data, but due to previous measurements on similar samples. Magnetostriction changes sign near to TSRT which led to the assumption that structural change at the morphotropic phase boundary (at TSRT) are responsible for change in magnetic properties and magnetostriction.

There are some points which are not clear and should be added/changed to help the reader:

  • Nothing is said about temperature at which X-ray measurements were made.
  • 1(b) is not very instructive. It should be exchanged by one where bonds and directions are shown.
  • Plots in Fig.2(a) are too small. One only can see the two transition temperatures, but has no chance to check if sample e.g. is paramagnetic above Tc. Here larger figures 1/Chi(T) should be shown; at least in the supplement.
  • Same for Fig.3(a). Enlargement of the inner part of the hysteresis loops should be added, to prove that there is a hysteresis and to be able to check HC.
  • In the discussion of Fig.3(b) it is argued that for x=0.6 and 0.7 MS is low and Hc is large, whereas for x=0.5 MS is high and HC is small, because x=0.5 is located farer away from MPB. Why this is not the case for x=0.3 and 0.4? Nothing is said about this. This has to be commented.
  • In Fig.4(a) for some temperatures the symbols have different color then the corresponding lines. This is confusing.
  • In line 154 it is said that TSRT for x=0.4 and 0.5 lie within the temperature range of 10 K ~ 130 K, but magnetostriction remains positive. Why x=0.3 is not mentioned? According to Fig.2(b) a TSRT is also present for x=0.3.
  • A table with numerical results would be good.
  • No comment is made, how high the obtained maximum magnetostriction is compared to other compounds. Is the figure of merit comparable to other compounds?
  • Page of citation [25] is wrong.

Reviewer 3 Report

The manuscript titled 'Magnetic and magnetostrictive behaviors of Laves-phase rare-earth - transition-metal compounds Tb1-xDyxCo1.95' by C. Zhou et al. reports on the preparation and characterization of in-title-mentioned Laves-phases. Subsequently, magnetization and magnetostriction measurements on prepared samples are presented and discussed in the frame of previous studies.

Although I do not find any real novelty in the submitted manuscript, in view of previous studies of similar alloys, I evaluate it as a solid study of the rare-earth - cobalt family with MPB. Therefore, I recommend to accept the manuscript for publication after following issues are addressed:

1) The choice of the system for a systematic study should be properly explained. What was the motivation?
2) The authors should discuss in detail their results on investigated alloys in the frame of the whole family. Are studied materials suitable for the applications? Is their potential higher than for previously studied alloys?
3) Language and typos should be corrected in the manuscript.
